# A New *Kayfunavirus*-like *Escherichia* Phage vB_EcoP-Ro45lw with Antimicrobial Potential of Shiga Toxin-Producing *Escherichia coli* O45 Strain

**DOI:** 10.3390/microorganisms11010077

**Published:** 2022-12-27

**Authors:** Xincheng Sun, Yen-Te Liao, Yujie Zhang, Alexandra Salvador, Kan-Ju Ho, Vivian C. H. Wu

**Affiliations:** 1Produce Safety and Microbiology Research Unit, U.S. Department of Agriculture, Agricultural Research Service, Western Regional Research Center, Albany, CA 94710, USA; 2College of Food and Biological Engineering, Zhengzhou University of Light Industry, Zhengzhou 450001, China; 3Henan Key Laboratory of Cold Chain Food Quality and Safety Control, Zhengzhou University of Light Industry, Zhengzhou 450001, China; 4Collaborative Innovation Center of Food Production and Safety, Zhengzhou 450001, China

**Keywords:** *Kayfunavirus*-like phage, STEC O45, lytic phage, genomic characterization, alternative antimicrobial agent

## Abstract

Lytic bacteriophages are re-considered as a solution to resolve antibiotic-resistant rampage. Despite frequent foodborne outbreaks caused by the top six non-O157 Shiga-toxin-producing *Escherichia coli* (STEC), the current interventions are not sufficiently effective against each serogroup, particularly O45. Therefore, this study aimed to characterize a new short-tailed phage, vB_EcoP-Ro45lw (or Ro45lw), as an alternative antimicrobial agent for STEC O45 strains. Phage Ro45lw belongs to the *Kayfunavirus* genus within the *Autographiviridae* family and shares no close evolutionary relationship with any reference phages. Ro45lw contains a tail structure composed of a unique tail fiber and tail tubular proteins A and B, likely to produce enzymatic activity against the target bacterial cells besides structural function. Additionally, the phage genome does not contain virulent, antibiotic-resistant, or lysogenic genes. The phage has a latent period of 15 min with an estimated burst size of 55 PFU/CFU and is stable at a wide range of pH (pH4 to pH11) and temperatures (30 °C to 60 °C). Regardless of the MOIs (MOI = 0.1, 1, and 10) used, Ro45lw has a strong antimicrobial activity against both environmental (*E. coli* O45:H-) and clinical (*E. coli* O45:H2) strains at 25 °C. These findings indicate that phage Ro45lw has antimicrobial potential in mitigating pathogenic STEC O45 strains.

## 1. Introduction

Shiga-toxin-producing *Escherichia coli* (STEC) strains, including O157 and the top six non-O157 serogroups, are renowned for producing Shiga toxin, capable of causing severe illnesses, such as hemolytic–uremic syndrome (HUS), and are implicated in many foodborne outbreaks in the United States and around the world [1]. STEC, in general, has contributed to approximately 265,000 illnesses, 3600 hospitalizations, and 30 deaths yearly [2]. Although the O157 serogroup is the primary pathogen contributing to STEC-associated outbreaks, the rate of non-O157 STEC infections has increased in recent years [3]. Among non-O157 STEC serogroups, STEC O45 strains have scarcely been implicated in any recent incidences but are still regulated as adulterants for certain types of food [4,5]. Despite the low prevalence of STEC O45 infection, precautions for this pathogen are still needed and can easily be overlooked. *E. coli* O45:H2 was associated with sporadic cases of bloody diarrhea from previous foodborne outbreaks, deriving from ill food workers and consuming contaminated meat products [4,6,7]. Our previous study also showed that *E. coli* O45:H2 of clinical or outbreak origin shared a close evolutionary relationship with the outbreak strain *E. coli* O103:H2, one of the most prevalent serogroups associated with recent foodborne outbreaks [8]. Most importantly, our genomic results showed that *E. coli* O45:H2 harbored similar pathogenicity factors as *E. coli* O103:H2 and could still cause severe illness if food producers or administrators overlook the precautionary interference to prevent human infection. A previous study found the genetic correlation of *E. coli* O104:H4 strain associated with a 2011 outbreak in Germany with the same strain, *E. coli* O104:H4 (HUSEC041), isolated 10 years ago in the same area where this particular serogroup of *E. coli* was not even within the radar [9]. The food industry has established standardized antimicrobial measurements, such as lactic acid and peroxyacetic acid, to prevent contamination of STEC O157 for years. However, these intervention technologies may not be sufficient to control each top six non-O157 STEC serogroups, and the efficacy varies depending on the contamination levels [10]. Therefore, an alternative option of antimicrobial interventions for these pathogens is needed.

Bacteriophages, also known as phages, are bacteria-killing viruses with high diversity and are ubiquitous in the ecosystem [11]. Phages can recognize their receptor proteins on the target bacterial cell walls to initiate phage infection [12]. For a lytic phage, phage DNA will then hijack the genomic machinery of the bacterial host to produce phage progenies, released via lysis of the bacterial cell [13]. Therefore, lytic phages are host-specific and are very helpful in treating bacterial infections, especially those with the development of antibiotic resistance, without affecting the biennial background flora [14]. Many studies have indicated the antimicrobial potential of lytic phages against various STEC strains, particularly O157 serogroup [15,16]. In contrast to chemical-based antimicrobial agents, the phage application has few adverse effects on the quality of food products, such as sprouts [17]. Consumer awareness of phage application in food is low; however, the Generally Recognized As Safe (GRAS) approved phage products shed new light on alternative antimicrobial agents for foodborne pathogens [18]. Thus, this study aimed to characterize *Escherichia* phage vB_EcoP-Ro45lw (or Ro45lw) isolated from non-fecal compost for the potential antimicrobial activity against STEC O45.

## 2. Materials and Methods

### 2.1. Phage Isolation

*Escherichia* phage Ro45lw was previously isolated with *E. coli* O45:H- from a non-fecal compost sample [19]. Phage propagation was performed by mixing 50 μL phage lysate (~10^6^ PFU/mL) with 100 μL of overnight *E. coli* O45:H- culture per plate using a double-layer plaque assay. After incubation at 37 °C for 20 h, 5 mL SM buffer was poured on top of the plate and then incubated at room temperature overnight to elute the phage. Later, the elution was centrifuged at 8000× *g* for 5 min, followed by filtration through a 0.22 μm filter membrane prior to downstream analysis. 

### 2.2. Bacterial Culture

A bacterial panel consisting of the top 6 non-O157 STEC—the serogroups O26, O45, O103, O111, O121, O145—*E. coli* O157:H7, generic *E. coli*, and *Salmonella enterica* strains was obtained from the Produce Safety and Microbiology (PSM) Research Unit at the U.S. Department of Agriculture (USDA), Agricultural Research Service (ARS), Western Regional Research Center (WRRC), Albany, CA, USA for the host range tests in this study. *E. coli* O45:H- was used for phage propagation, quantification, and the one-step growth curve. Fresh bacterial culture was prepared in 10 mL tryptic soy broth (TSB; Becton Dickinson, Sparks, MD, USA) inoculated with 1 μL loopful of individual strain and incubated overnight at 37 °C before use.

### 2.3. Genomic Analysis

Phage Ro45lw was purified through a CsCl gradient and subsequently subjected to DNA extraction, followed by DNA library preparation before sequencing as previously described [20]. Later, the samples were loaded into a MiSeq Reagent Kit v2 (500-cycle) and sequenced on MiSeq platform (Illumina, San Diego, CA, USA), generating approximately 14 million 2 × 250 bp paired-end reads. The updated assembly and annotation were generated using the previous methods [21] and deposited in National Center for Biotechnology Information (NCBI) database. Briefly, raw sequence reads were subjected to FASTQC and trimming via Trimmomatic with a setting of Q30. De novo assembly was conducted on the resulting quality reads using Unicycler Galaxy v0.4.6.0 (SPAdes v2.5.1), followed by annotation via the Prokka pipeline Galaxy 1.13 [22] with default settings. Subsequently, the annotation was manually curated with Universal Protein Resource (UniProt) database [23] using Geneious (v11.0.3, Biomatters, Auckland, New Zealand). tRNAscan-SE (v2.0) server was used to confirm the predicted tRNAs in the phage genome [24]. Genome map of Ro45lw was performed using the CGview server beta (https://proksee.ca/, accessed on 15 November 2022).

The evolutionary tree of Ro45lw with its close-related reference phages was analyzed using the Virus Classification and Tree Building Online Resource (VICTOR) based on the whole-genome sequence at the amino acid level [25]. Phylogenetic analyses of the amino acid sequences were conducted on the putative tail fiber protein (ORF_44), tail tubular protein B (ORF_37), lysozyme (ORF_17), and putative class II holin (ORF_45) using the methods as previously described [20]. In brief, the amino acid sequences were aligned using the ClustalW (version 1.2.3), and the phylogenetic tree was performed using the MEGA11 program, with the maximum composite likelihood method and 500 bootstrap replications.

### 2.4. Transmission Electron Microscopy (TEM)

Purified phage Ro45lw was CsCl-purified before being subjected to negative staining and examined in a transmitted electron microscope (FEI Tecnai G2), as previously described [20].

### 2.5. One-Step Growth Curve

A one-step growth curve of phage Ro45lw was conducted on *E. coli* O45:H1-, based on the previous method with subtle modification [20]. In brief, fresh bacterial culture was prepared in 10 mL of TSB at 37 °C for 20 h; the next day, 0.2 mL of the overnight culture was sub-cultured in 19.8 mL of TSB and incubated for 2 h at 37 °C to reach log phase of the bacterial growth. Thereafter, phage lysate of Ro45lw was added to the log-phase bacterial solution (MOI of 0.01) supplemented with CaCl_2_ at 10 mM and incubated at 37 °C for 5 min to allow phage adsorption onto the membranes of the bacterial cells. The phage–bacterial mixture was centrifuged at 10,000× *g* for 5 min at 4 °C to discard supernatant. Later, the bacterial pellet was washed with 2 mL of TSB and then resuspended into 20 mL of fresh TSB. Next, 0.3 mL of resuspended culture was added to 29.7 mL of TSB and incubated at 37 °C throughout the duration of the experiment (35 min). Meanwhile, quantification of phage-infected bacteria was determined at incubation time 0 by mixing 50 μL of the 30 mL phage–bacterial mixture (no filtration) with 100 μL of fresh overnight bacterial culture and 5 mL of molten 50% tryptic soy agar (TSA) and pouring into a Petri dish with 10 mL bottom TSA (also known as double-layer plaque assay). Additionally, 1 mL of the sample was obtained from the 30 mL phage–bacterial mixture every five min and subjected to 0.22-µm membrane filtration. The titers of Ro45lw were determined using the double-layer plaque assay, where plates were incubated at 37 °C overnight. The experiment was conducted in three replications to estimate the latent period and burst size of phage Ro45lw.

### 2.6. pH and Temperature Stability

A range of pH values from pH3 to pH12 was used to test the stability of Ro45lw at 37 °C for 20 h using the method as previously described with minor changes [20]. In brief, 100 μL of phage Ro45lw was added to 5 mL of SM buffer with final pH values of 3, 4, 5, 7, 10, and 12 and incubated at 37 °C for 20 h. Viable phage particles of Ro45lw were determined against **E. coli* O45*:H- via the double-layer plaque assay. 

For the temperature stability test, bulk phage lysate of Ro45lw was prepared by mixing the original phage lysate with SM buffer (pH = 7) at 1:9 (*v*/*v*) ratio. Later, an aliquot of 1 mL Ro45lw solution was dispensed in several sterile microcentrifuge tubes and subjected to heat treatment, including 30, 40, 50, 60, and 70 °C, for 20 min, 40 min, and 60 min. The temperatures covered were inclusive of the general conditions encountered in food-related environments. Phage titers were determined using the double-layer plaque assay. 

### 2.7. Host Range and Efficiency of Plating (EOP)

The host range test of Ro45lw against non-pathogenic *E. coli*, *E. coli* O157:H7, top six non-O157 STEC, and three *Salmonella* strains was conducted using a spot test assay with three different dilutions of phage lysate (10^−1^ to 10^−3^) as previously described [26]. The bacterial strains showing lysis were further subjected to the efficiency of plating (EOP) assay to determine productive infection of phage Ro45lw from the tested bacterial strains versus the primary host strain for producing the phage progenies [27]. Briefly, fresh bacterial cultures were prepared in TSB at 37 °C overnight and used for quantification of Ro45lw by the double-layer plaque assay with diluted phage lysate using four successive dilutions (10^−3^ to 10^−7^). The plates were incubated at 37 °C for 18 h. The experiment was conducted in three replications. Generally, a high phage-producing efficiency had an EOP of 0.5 or more; a medium-producing efficiency had an EOP above 0.1 but below 0.5; a low-producing efficiency had an EOP between 0.001 and 0.1; inefficient phage production was any value lower than 0.001.

### 2.8. Bacterial Challenge Assay

The bacterial challenge assay was performed to measure the effects of phage Ro45lw with different MOIs on bacterial growth based on bacterial optical density at a wavelength of 600 nm (OD_600_) as previously described with minor changes [21]. In brief, fresh bacterial culture of *E. coli* O45:H- (RM10729), *E. coli* O45:H16 (RM13752), and *E. coli* O45:H2 (SJ7) was prepared in TSB at 37 °C overnight and further diluted in TSB to 1 × 10^4^ CFU/mL. An aliquot of 190 µL of diluted bacterial solution per well was added to a 96-well plate. Later, 10 µL of phage Ro45lw with different titers was added to each sample well to reach MOIs of 0.05, 0.5, and 5; the control group contained only bacterial solution without phage. At 25 °C, readings programmed to OD_600_ were measured using a spectrophotometer (Promega, Madison, WI, USA) every 30 min for 17 h. The experiment was conducted in 3 replications.

### 2.9. Statistical Analysis

Experiments subjected to statistical analysis were conducted in three individual repeats. The quantification of phages was calculated as PFU/mL, with logarithmical conversion for statistical analysis. The stress effect of pH (pH4 to pH10) was determined using a one-way analysis of variance (ANOVA) with statistical significance at a 5% level [28].

## 3. Results

### 3.1. Genomic Features of Ro45lw

Phage Ro45lw had double-stranded DNA with 39,793 bp genome size and an average GC content of 52.2%. The phage was predicted to have the packaging mechanism belonging to direct terminal repeats, also known as DTR [29], with 190 bp terminal repeat region at both sides of the genome end [19]. Ro45lw also contained 50 open reading frames (ORFs), 6 promoters, and 1 terminator without tRNA (Figure 1). Ro45lw had been classified in the genus of *Kayfunavirus* under the *Autographiviridae* family by the International Committee on Taxonomy of Virus (ICTV). Based on the ICTV taxonomy guideline, a new phage classified in the same genus shares a minimum 70% nucleotide identity of the full genome length of the phages in the particular genera via analysis of genomic tools, such as BLASTn or VIRIDIC [30]. The evolutionary tree of the VICTOR analysis showed that Ro45lw had a unique amino acid sequence in comparison to 65 reference phages belonging to the *Kayfunavirus* genus (Figure 2). Among 50 ORFs, 26 were annotated with known functions, including structural proteins, DNA replication, DNA packaging, and cell lysis (Table 1). The phage also contained regulatory genes, such as phi1.6, phi2.5, phi6.5, and phi17, which were also found in Enterobacteria phage T7.

#### Phylogenetic Analysis

Phylogenetic analyses were conducted on the ORFs encoding the proteins associated with host range and bacterial cell lysis. Ro45lw contained a unique putative tail fiber protein sharing no close evolutionary relationship at the amino acid level with the counterfeit of any reference phage (Figure 3A). Moreover, Ro45lw contained ORF_37 encoding tail tubular protein B, which shared a close evolutionary relationship with Cronobacter phage GW1 (Figure 3B). The phage also harbored the gene encoding tail tubular protein A, with high amino acid similarity to the counterpart of Cronobacter phage GW1 (data not shown). Tail fiber, tail tubular proteins A and B are parts of the tail structure of a podophage [31]. For lysozyme (cell lysis), Ro45lw shared a high amino acid identity with the counterpart of *Escherichia* phage IMM-002 (Figure 3C). On the contrary, the phage contained putative class II holin protein that did not share a close evolutionary relationship with that of the reference phages (Figure 3D).

### 3.2. Phage Morphology

Phage Ro45lw had the morphology containing an icosahedral head of approximately 95 ± 5 nm in diameter and a short tail, showing the *Podoviridae* morphology (Figure 4A). The phage also produced plaques with a distinct halo on the plaque assay plate (Figure 4B).

### 3.3. One-Step Growth Curve and Stability Test

Phage Ro45lw had a 15 min latent period against *E. coli* O45:H- (RM10729) with an average burst size of 55 progeny virions per infected cell after the incubation at 37 °C for 35 min (Figure 5).

For pH tolerance, the titers of phage Ro45lw dropped approximately 2 log PFU/mL (*p* < 0.05) at pH4 and 1 log PFU/mL (*p* < 0.05) at pH5, pH10, and pH11 after 20 h incubation at 37 °C (Figure 6A). No phage was detected at either pH3 or pH12. For temperature stability, Ro45lw was stable at temperatures ranging from 30 °C to 50 °C for 1 h treatment but completely inactivated at 60 °C and 70 °C after 40 min and 20 min, respectively (Figure 6B).

### 3.4. Host Range and Productive Infection

Phage Ro45lw was able to infect various STEC O45 strains of environmental and clinical origins and generic *E. coli* (ATCC 13706 & DH5α) strains; no *Salmonella* strains were subjected to the phage infection (Table 2). Except for the generic *E. coli* with inefficiency, both environmental *E. coli* O45:H16 strains had a high phage-producing efficiency, and one clinical *E. coli* O45:H2 possessed a medium phage-producing efficiency (Table 2).

### 3.5. Bacterial Challenge Assay of Ro45lw

The in vitro antimicrobial activities of Ro45lw at MOIs of 0.1, 1, and 10 were determined against environmental *E. coli* O45:H16 (RM13752) and clinical *E. coli* O45:H2 (SJ7) strains at 25 °C using a spectrophotometer (Figure 7). The results showed that the bacterial growth of both *E. coli* O45:H16 and *E. coli* O45:H6 strains were completely inhibited by Ro45lw regardless of MOIs used during the 17.5 h incubation (Figure 7). 

## 4. Discussion

Phage Ro45lw is a new member of podophages with a short tail, showing genomic classification in the genus of *Kayfunavirus* within the *Autographiviridae* family. Ro45lw does not share a close evolutionary relationship at the amino acid level with any of the 65 reference phages used in this study which also aligns with the VICTOR classification analysis, showing no similarity to Ro45lw at the species level. The Ro45lw does not contain tRNA in the phage genome, which could be associated with a narrow host range of the phage [32]. This is related to high codon usage by the phages containing more tRNAs than those with few or none [32]. 

For podophages, T7-like phages in particular, the tail structure is composed of at least four proteins, including connector protein, tail tubular proteins, and fiber proteins [31]. Previous studies found that tail tubular protein A was associated with phage adherence to its bacterial hosts, with hydrolytic activity targeting polysaccharides on the bacterial capsule [33,34]. However, the enzymatic function of tail tubular protein A, also known as depolymerase enzyme [35], likely contributes to the halo surrounding a plaque caused by phage Ro45lw. The same research team also found the additional enzymatic activity of tail tubular protein B, extracted from *Yersinia* phage, to inhibit bacterial growth and biofilm formation of *Yersinia enterocolitica*, besides structural function [36]. Due to the close evolutionary relationship to *Cronobacter* phage GW1 on both tail tabular protein A and B, phage Ro45lw might be able to infect *Cronobacter* commonly found in dry food, such as powdered milk [37]. The current results show that Ro45lw has a unique tail fiber protein sequence. The phage also encodes a lysozyme sharing high amino acid similarity to *Escherichia* phage IMM-002, which infects multiple STEC O6 strains [38]. This protein likely contributes to a strong affinity toward STEC strains because its C-terminal domain is related to recognizing and adsorbing bacterial lipopolysaccharide on the cell membrane [39]. The special tail structure and cell lysis enzyme likely render the antimicrobial activity of Ro45lw against clinical *E. coli* O45:H2 strains, which were resistant to the infection of another STEC O45-infecting phage, Sa45lw, previously isolated in our lab [20]. Furthermore, Ro45lw can suppress both environmental and clinical STEC O45 with various MOIs in this study. The genome of Ro45lw does not have virulent genes, any lysogenic genes, or any antibiotic-resistance genes that can jeopardize the phage application as an alternative antimicrobial agent. 

Regarding growth parameters, Ro45lw has a short latent period of 15 min and a small burst size (55 PFU/CFU) against *E. coli* O45:H-. Ro45lw encodes a unique class II holin protein, one of the factors affecting the latent period of the phage [40]. Ro45lw contains transcriptional terminators and regulated genes found in T7 phage and may be associated with the phage’s lysis time [41]. Despite Ro45lw having a small burst size and short latent period, it may rapidly establish multiple parallel infections on the target bacteria if the bacterial concentration is high [42]. Therefore, the initial titers of the phage may be more critical upon application than the subsequent infection by the phage progenies produced from the first infection to mitigate the target bacteria. The current results showed that Ro45lw is stable at a wide pH range but is more sensitive to thermal treatment than phage Sa45lw, which could sustain up to 65 °C for 1 h [20]. The overall phage stability of Ro45lw is adequate to support further various applications in the food-processing environment.

For antimicrobial activity, although Ro45lw has a narrow host range and a small burst size compared with the phage Sa45lw [20], the phage characterized in this study can infect STEC O45 of both environmental and clinical sources. The finding indicates that Ro45lw can target different receptor proteins on these STEC O45 strains from phage Sa45lw, and both phages may be used for developing a phage cocktail to enhance antimicrobial activity. Further study regarding in vitro bacterial reduction and phage application in a food model is necessary.

To conclude, the genetic evidence indicates that phage Ro45lw is a novel STEC O45-infecting phage in the *Kayfunavirus* genus. The current findings show that phage Ro45lw has the antimicrobial potential to control pathogenic STEC O45 strains. Additionally, the phage-derived depolymerases with enzymatic activity encoded by Ro45lw may also be promising phage-derived antimicrobial agents to prevent bacterial pathogens. 

## Figures and Tables

**Figure 1 microorganisms-11-00077-f001:**
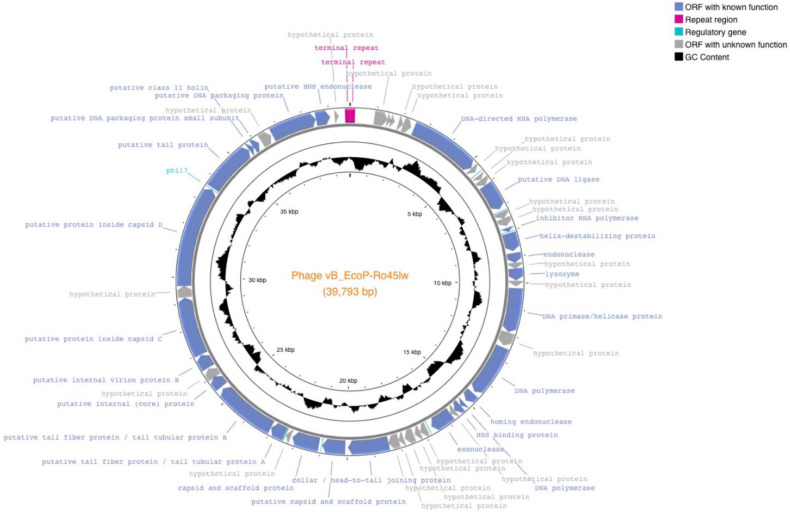
Genome map of phage Ro45lw was generated using the CGview server beta. ORFs are annotated and colored based on predicted molecular functions of identified genes (Table 1). The center of the genome map provides % GC content (black).

**Figure 2 microorganisms-11-00077-f002:**
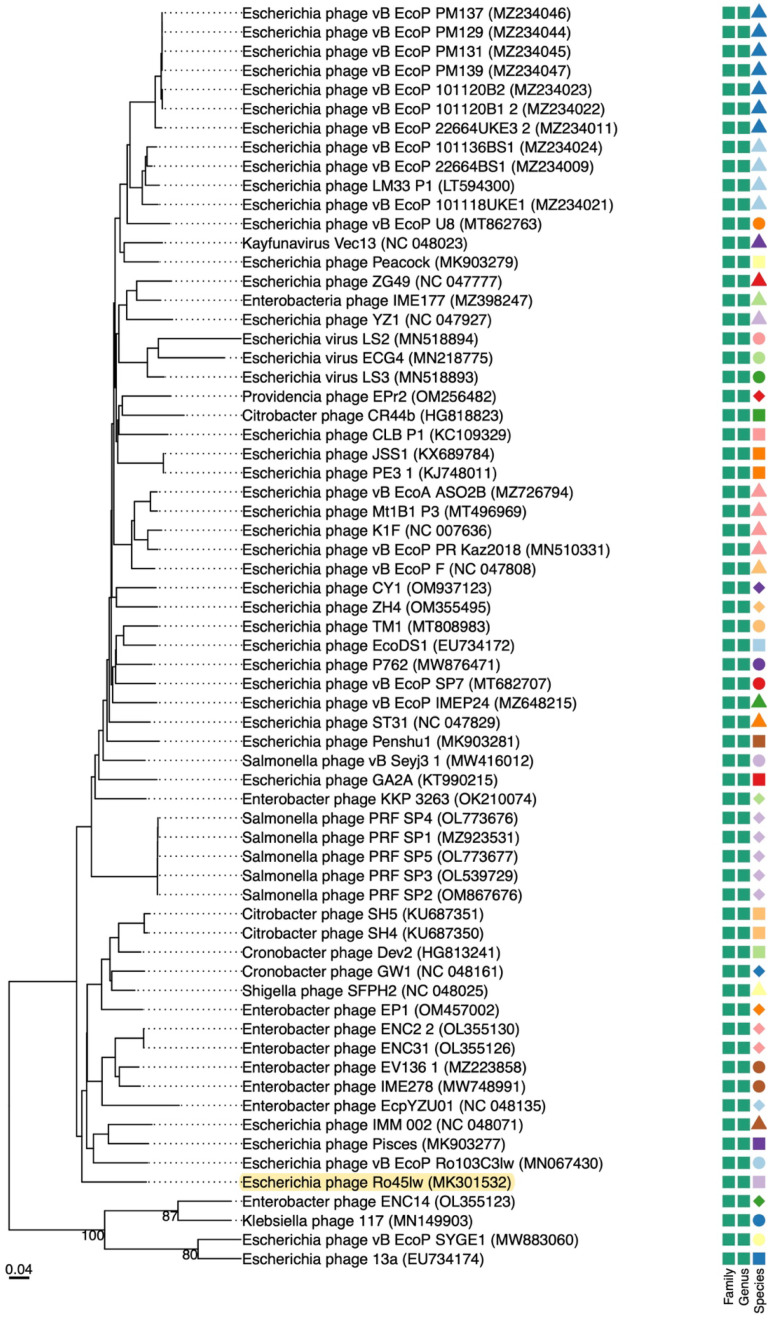
Phylogenetic analysis of whole-genome sequences of Ro45lw and its closely related reference phages belonging to the *Kayfunavirus* genus under the *Autographiviridae* family at the amino acid level using VICTOR (formula d6). Annotations, including family, genus, and species cluster predicted by VICTOR, with different shapes and colors, are different in classification.

**Figure 3 microorganisms-11-00077-f003:**
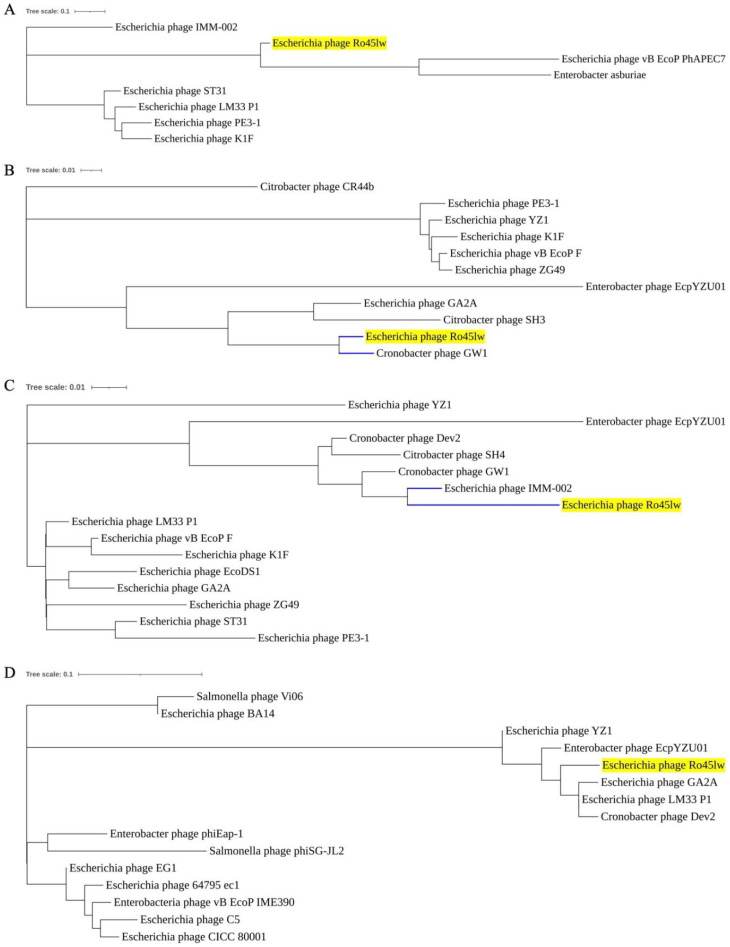
Phylogenetic tree of phage Ro45lw (with yellow highlight) and the close-related reference phages based on the Clustal Omega alignment of the amino acid sequences of putative tail fiber protein (ORF_44) (**A**), tail tubular protein B (ORF_37) (**B**), lysozyme (ORF_17) (**C**), and putative class II holin (ORF_45) (**D**). Blue lines are used to indicate the closest evolutionary relationship. The scale represents the homology percentage.

**Figure 4 microorganisms-11-00077-f004:**
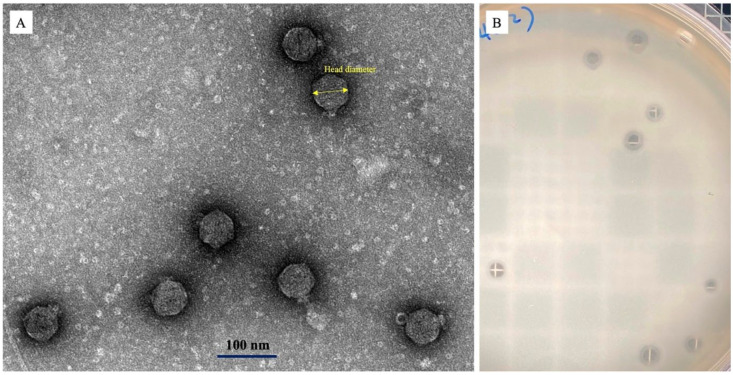
Phage morphology under (**A**) transmission electron microscopy with a capsid (95 ± 5 nm in diameter) and a short tail, and (**B**) plaque morphology on a plaque-assay plate.

**Figure 5 microorganisms-11-00077-f005:**
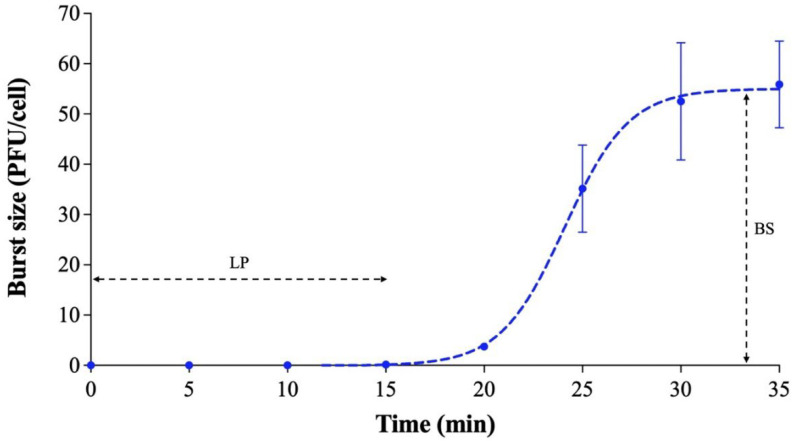
One-step growth curve of phage Ro45lw using generic *E. coli* O45:H- strain (RM10729). The growth parameters of the phage indicate a latent period (LP) of 15 min and an average burst size (BS) of 55 phages per infected cell. The error bars present the standard error of the mean (SEM) for each time point of the one-step growth curve.

**Figure 6 microorganisms-11-00077-f006:**
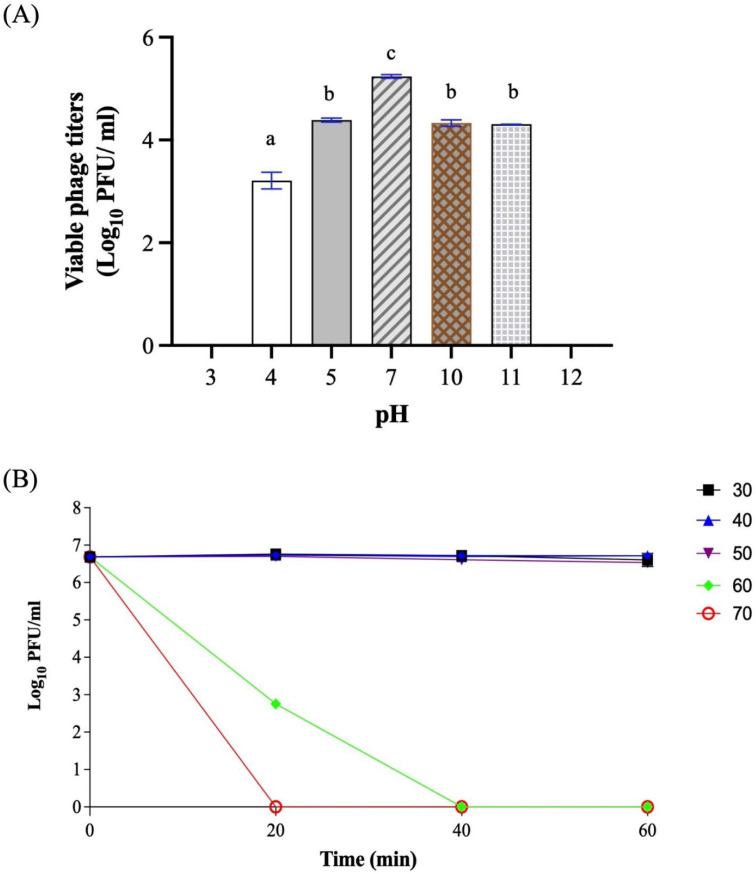
Stability of phage Ro45lw at (**A**) different pH levels (pH 3, pH 5, pH 7.5, pH 9, pH 10.5, and pH 12) at 37 °C for 20 h, and (**B**) various temperatures (30 °C to 70 °C) for up to 1 h. For the pH test, the means of phage titers that lack common letters (a, b, and c) differ (*p* < 0.05). The error bars show the SEM.

**Figure 7 microorganisms-11-00077-f007:**
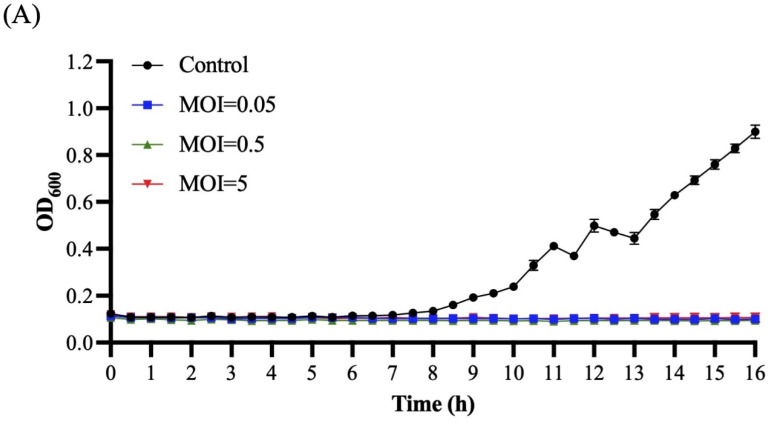
Bacterial challenge assay for *E. coli* O45:H16 (**A**) and *E. coli* O45:H2 (**B**) treated with phage Ro45lw at MOIs of 0.1, 1 and 10 at 25 °C for 17.5 h based on the bacterial optical density at wavelength 600 nm (OD_600_). The Control group only contains bacterial culture.

**Table 1 microorganisms-11-00077-t001:** List of annotated ORFs with known predicted functions, size, and location in the Ro45lw genome.

Type	Predicted Function	Functional Category	Minimum bp	Max bp
Repeat region	terminal repeat region	-	1	190
ORF_6	DNA-directed RNA polymerase CDS	DNA replication	2498	5179
regulatory	predicted promoter regulatory	-	5483	5505
ORF_10	putative DNA ligase CDS	DNA replication	6026	7093
regulatory	phi1.6 regulatory	-	7178	7200
ORF_13	inhibitor RNA polymerase CDS	DNA replication	7864	8022
regulatory	phi2.5 regulatory	-	8018	8040
ORF_14	helix-destabilizing protein CDS	DNA replication	8080	8778
ORF_15	endonuclease CDS	DNA replication	8859	9233
ORF_17	lysozyme CDS	Cell lysis	9450	9908
ORF_19	DNA primase/helicase protein CDS	DNA replication	10,203	11,903
ORF_21	DNA polymerase CDS	DNA replication	12,523	14,541
ORF_22	homing endonuclease CDS	DNA replication	14,621	15,016
ORF_23	DNA polymerase CDS	DNA replication	15,119	15,292
ORF_24	HNS binding protein CDS	DNA replication	15,292	15,582
ORF_26	exonuclease CDS	DNA replication	15,785	16,651
regulatory	phi6.5 regulatory	-	16,774	16,796
ORF_32	collar/head-to-tail joining protein CDS	Structural protein	18,416	19,984
ORF_33	putative capsid and scaffold protein CDS	Structural protein	20,081	20,962
regulatory	phi regulatory	-	20,967	20,989
ORF_34	capsid and scaffold protein CDS	Structural protein	21,093	22,142
regulatory	phiTE regulatory	-	22,409	22,444
ORF_36	putative tail fiber protein/tail tubular protein A CDS	Structural protein	22,464	23,030
ORF_37	putative tail fiber protein/tail tubular protein B CDS	Structural protein	23,042	25,411
ORF_38	putative internal (core) protein CDS	Structural protein	25,498	25,980
ORF_40	putative internal virion protein B CDS	Structural protein	26,381	26,944
ORF_41	putative protein inside capsid C CDS	Structural protein	26,956	29,238
ORF_43	putative protein inside capsid D CDS	Structural protein	29,675	33,571
regulatory	phi17 regulatory	-	33,567	33,589
ORF_44	putative tail protein	Structural protein	33,637	35,646
ORF_45	putative class II holin CDS	Cell lysis	35,662	35,856
ORF_46	putative DNA packaging protein small subunit CDS	DNA packaging	35,853	36,116
ORF_48	putative DNA packaging protein CDS	DNA packaging	36,710	38,476
ORF_49	putative HNH endonuclease CDS	DNA replication	38,486	39,028
Repeat region	terminal repeat region	-	39,604	39,793

**Table 2 microorganisms-11-00077-t002:** Host range and efficiency of plating of phage Ro45lw against various serogroups of Shiga-toxin-producing *E. coli* and *Salmonella enterica* strains.

Group	Strain ID	EOP ^α^
Non-O157 STEC	STEC O26, O103, O111, O121 and O145	R *
	*E. coli* O45:H- (RM10729)	H ^
	*E. coli* O45:H2 (SJ7)	0.10
	*E. coli* O45:H16 (RM13752)	0.77
	*E. coli* O45:H16 (RM13745)	0.84
STEC O157	*E. coli* O157:H7 (RM18959, RM18974 & ATCC 43888)	R
Generic *E. coli*	ATCC 13706	Inefficiency
	DH5a	R
*Salmonella enterica*	*Salmonella* Newport	R
	*Salmonella* Enteritidis	R
	*Salmonella* Typhimurium	R

^α^ EOP was conducted for the spot test-positive strains, with a value calculated by the ratio of phage titer on the test bacterium versus the phage titer on the primary host bacterium. High production efficiency is EOP ≥ 0.5, medium production efficiency is 0.5 > EOP ≥ 0.1, low production efficiency is 0.1 > EOP > 0.001, and inefficiency of phage production is EOP ≤ 0.001. * R denotes no lysis in the spot test assay. ^ H was the primary host bacterial strain used for isolation.

## Data Availability

The phage Ro45lw sequence used in this study can be found in GenBank, with the accession number MK301532.

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
