# Peer review of "A New Kayfunavirus-like Escherichia Phage vB_EcoP-Ro45lw with Antimicrobial Potential of Shiga Toxin-Producing Escherichia coli O45 Strain"

_microorganisms, 2022, doi:10.3390/microorganisms11010077_

Round 1

Reviewer 1 Report

excellent work as always,  I thank the authors for their great research.  few comments to consider:

we need a reference for the statistical analysis.

the conclusion is very short and does not provide details as related to the objective.   Include limitations of your work and possible future steps.   good work... 

Author Response

Comments and Suggestions for Author 1:

excellent work as always, I thank the authors for their great research.  few comments to consider:

we need a reference for the statistical analysis.

Response:

The related reference is included in the updated manuscript.

the conclusion is very short and does not provide details as related to the objective.   Include limitations of your work and possible future steps.   good work... 

Response:

More information regarding antimicrobial activity is included in the discussion of the updated manuscript.

Reviewer 2 Report

Authors did a great job to characterize a new Kayfunavirus-like Escherichia phage vB_EcoP-Ro45lw for the antimicrobial potential of Shiga toxin-producing E. coli O45 strain. The main concern is how Ro45lw was classified in the genus of Kayfunavirus under the Autographiviridae family.

“Ro45lw had been classified in the genus of Kayfunavirus under the Autographiviridae family by International 184 Committee on Taxonomy of Virus (ICTV)” line 183-184

How was this classification (genus) achieved? Please check the guidelines on ICTV taxonomy by Turner et al 2021. I think the Virus Intergenomic Distance Calculator (VIRIDIC) will guide the classification. Moraru et al, 2020. This will verify the nucleotide similarity of Phage Ro45lw with closely related phages and based on threshold for genera (>70%) and specie (>95%), the genus can be determined.

Minor concerns

"Although the food industry has established standardized antimicrobial measurements to prevent contamination of STEC O157 for years," please expand on this by giving examples

“For the temperature stability test, bulk phage lysate of Ro45lw was prepared by mixing the original phage lysate with SM buffer at 1:9 (v/v) ratio” Line 140: Please indicate SM buffer pH.

Please check spacing "overlook/ ignore " line 52

Pleases italicized E. coli in line 250-254 and throughout the text

Please check this 37ËšC and 37 ËšC in line 153 and 155. Which one is correct?

Author Response

Comments and Suggestions for Author 2

Authors did a great job to characterize a new Kayfunavirus-like Escherichia phage vB_EcoP-Ro45lw for the antimicrobial potential of Shiga toxin-producing E. coli O45 strain. The main concern is how Ro45lw was classified in the genus of Kayfunavirus under the Autographiviridae family.

“Ro45lw had been classified in the genus of Kayfunavirus under the Autographiviridae family by International 184 Committee on Taxonomy of Virus (ICTV)” line 183-184

How was this classification (genus) achieved? Please check the guidelines on ICTV taxonomy by Turner et al 2021. I think the Virus Intergenomic Distance Calculator (VIRIDIC) will guide the classification. Moraru et al, 2020. This will verify the nucleotide similarity of Phage Ro45lw with closely related phages and based on threshold for genera (>70%) and specie (>95%), the genus can be determined.

Minor concerns

"Although the food industry has established standardized antimicrobial measurements to prevent contamination of STEC O157 for years," please expand on this by giving examples

Response:

The revised information is included in the updated manuscript.

“For the temperature stability test, bulk phage lysate of Ro45lw was prepared by mixing the original phage lysate with SM buffer at 1:9 (v/v) ratio” Line 140: Please indicate SM buffer pH.

Response:

The information has been provided in the updated manuscript.

Please check spacing "overlook/ ignore " line 52

Response:

We have removed “/ ignore” in the updated manuscript.

Pleases italicized E. coli in line 250-254 and throughout the text

Response:

The revision is included in the updated manuscript.

Please check this 37ËšC and 37 ËšC in line 153 and 155. Which one is correct?

Response:

We used “space” between the number and unit; the revision has been included in the updated manuscript.